# Leveraging Artificial Intelligence and Provenance Blockchain Framework to Mitigate Risks in Cloud Manufacturing in Industry 4.0

**Mifta Ahmed Umer** [1,*] **, Elefelious Getachew Belay** [1] **and Luis Borges Gouveia** [2]

[1] School of Information Technology and Engineering, Addis Ababa Institute of Technology (AAiT), Addis Ababa University, Addis Ababa 18869, Ethiopia; elefelious.getachew@aait.edu.et
[2] Science and Technology Faculty, University Fernando Pessoa, 349 4249-004 Porto, Portugal; lmbg@ufp.edu.pt
* Correspondence: mifsat@gmail.com

**Abstract:** Cloud manufacturing is an evolving networked framework that enables multiple manufacturers to collaborate in providing a range of services, including design, development, production, and post-sales support. The framework operates on an integrated platform encompassing a range of Industry 4.0 technologies, such as Industrial Internet of Things (IIoT) devices, cloud computing, Internet communication, big data analytics, artificial intelligence, and blockchains. The connectivity of industrial equipment and robots to the Internet opens cloud manufacturing to the massive attack risk of cybersecurity and cyber crime threats caused by external and internal attackers. The impacts can be severe because the physical infrastructure of industries is at stake. One potential method to deter such attacks involves utilizing blockchain and artificial intelligence to track the provenance of IIoT devices. This research explores a practical approach to achieve this by gathering provenance data associated with operational constraints defined in smart contracts and identifying deviations from these constraints through predictive auditing using artificial intelligence. A software architecture comprising IIoT communications to machine learning for comparing the latest data with predictive auditing outcomes and logging appropriate risks was designed, developed, and tested. The state changes in the smart ledger of smart contracts were linked with the risks so that the blockchain peers can detect high deviations and take actions in a timely manner. The research defined the constraints related to physical boundaries and weightlifting limits allocated to three forklifts and showcased the mechanisms of detecting risks of breaking these constraints with the help of artificial intelligence. It also demonstrated state change rejections by blockchains at medium and high-risk levels. This study followed software development in Java 8 using JDK 8, CORDA blockchain framework, and Weka package for random forest machine learning. As a result of this, the model, along with its design and implementation, has the potential to enhance efficiency and productivity, foster greater trust and transparency in the manufacturing process, boost risk management, strengthen cybersecurity, and advance sustainability efforts.

**Keywords:** provenance; blockchain; smart contract; predictive auditing; cloud manufacturing risks; industrial internet of things

## 1. Introduction

Cloud manufacturing, as the name suggests, is a framework of operational planning, scheduling, monitoring, and control of manufacturing operations using hosted applications through cloud computing [1–3]. Traditional manufacturing systems were controlled by programmable logic controllers (PLCs) operated by the local on-plant computers, which could run manufacturing operations in limited physical spaces. The software systems used for materials planning, operations scheduling, monitoring, and control were also hosted within the data centers of the manufacturing plants. These systems were not connected to the Internet as they were networked using proprietary protocols and connections.

Hence, the manufacturing operations were not exposed to cyber security threats in their traditional designs. To develop dynamic capabilities to respond to rapid demand and supply changes, manufacturers entered strategic alliances to cover larger customer bases and meet demands during normal times as well as during uncertainties and shocks [4–6]. For collective operations, manufacturers needed to integrate their operations, which was possible by creating digital values using digitalization systems in Industry 4.0 technologies [7,8]. The digital value proposition could be achieved by making the PLCs operate with open communication protocols by transforming them into cyber-physical systems. The newly evolved Industrial Internet of Things (IIoTs) was used to transform the PLCs into cyber communication devices that could interface with the Internet and be controlled remotely. With this technological development and the already recognized problem of disconnected manufacturing silos and crunch of resources in computing, memory, and storage capacities of the data centers operated and maintained by manufacturing organizations, cloud manufacturing became a viable option for the future of manufacturing. However, connectivity to the Internet opened the gate for cybersecurity threats to cloud manufacturing; these had not been a factor in general manufacturing environments as they had not been connected to the Internet [9]. Cloud-based PLCs were vulnerable to the same cybersecurity threats that caused vulnerabilities on the Internet-exposed computing systems. In general manufacturing, the PLCs were shielded from all such threats as their computing systems were not Internet-exposed.

Interfacing PLCs with the Internet in order to control them through cloud manufacturing applications hosted on cloud computing opened the gate for cyber threats to manufacturing organizations [9]. Cyber security threats are already prominent in the manufacturing industry. By 2017, about 75% of oil and gas industry businesses had suffered at least one successful attack causing measurable business impacts. Power grids have suffered about 15% of the total number of cyber attacks in 2017. More recent statistics reported by the Varonis and Forbes websites [10,11] reflect the ongoing trends of cyber attacks on manufacturing systems. Their reports stated that malicious power shell scripts targeted at cyber physical devices (detected and blocked) increased by 1000% to about 5200 monthly average attacks in 2021 and 2022 [10,11]. Normally, protection against remote code execution tactics is robust but rogue IIoT devices installed by insiders can cause a major loophole, especially by using malicious and non-transparent algorithms [12–14]. The more worrying trend is about insiders creating deliberate loopholes in the cyber physical systems of manufacturing plants, thus opening an attack surface for external exploits to penetrate and use the compromised cyber physical systems as launch pads [15–17]. The activity is reported to be about 30% of the overall number of attacks [10,11]. The extent to which unsolicited IIoT devices can be sneaked into manufacturing networks has not yet been estimated. However, 30% of the 5200 cyber attacks on IIoT devices in 2021 and 2022 were carried out through insider activity, which suggests a significant trend that is expected to increase [10,11]. In order to address these challenges, cyber security threats need to be visualized with a different perspective as compared with those threats in self-hosted manufacturing and supply chain computerized control systems [17].

This research presents design, prototyping, and testing of controls employing Artificial Intelligence (AI) and Provenance Blockchain framework for protecting organizations using cloud manufacturing applications against cyber security risks. As these organizations are having their PLCs transformed to cyber physical systems, they should be certified and accounted for at the time of inception and during their operations. As shown in the literature review, provenance is the dynamic metadata of systems and devices that captures their "data about their manipulation history" (including change of ownership and assignment to various roles). As further reviewed in literature review, blockchains can be used to form trusted networks of partners operating their assets in supply chains and logistics to serve common demands and orders. It is hereby emphasized that if the provenance system can be deployed on such blockchains to capture the "real-time operational data about the manipulation history" of cyber physical systems used in logistics

and supply chain operations for cloud manufacturers, it can help in mitigating cyber security risks to them by conducting AI-enabled predictive auditing. The research questions of this study are as follows:

1.  What are the risks associated with cloud manufacturing in Industry 4.0?
2.  How can provenance blockchain be used to provide greater transparency and traceability in the cloud manufacturing process using AI-enabled predictive auditing?
3.  How can this system help in mitigating cloud manufacturing risks in Industry 4.0?

In this study, the first research question is answered through literature review, the second research question is answered through designing, developing, and testing a software prototype, and the third research question is answered through a critical analysis of the software prototype, keeping in context the findings in the answer to the first research question. The next section presents a review of the literature.

## 2. Literature Review

The modern era for the manufacturing sector is highly competitive, dynamic, and complex, with uncertainties beyond the controls of individual companies [4,5]. To compete, survive, and flourish in this environment, manufacturing organizations need to develop "dynamic capabilities" to manage rapid changes as per the demand and competitive dynamics of their target markets [4–6]. Building dynamic capabilities requires strategic alliances among multiple manufacturing organizations and the use of modern technology to develop incremental improvements and rapid adjustments of manufacturing resources, processes, knowledge, and skills through management controls. The strategic alliances can be executed by creating a joint cloud manufacturing portal of applications that can monitor and control the manufacturing processes of the plants of the collaborating companies in the strategic alliance [1–3]. Industry 4.0 technologies and processes are viewed as the foundation for developing dynamic capabilities for cloud manufacturing [3,4,7,8]. Industry 4.0 technologies and processes have an influence on digitalization, digital value creation, real time knowledge of markets and demands, quick production and marketing, the ability to use and reuse materials and resources optimally, and sustainable development [4].

As stated in the introduction, cloud manufacturing comprises PLCs transformed into cyber physical systems running the manufacturing controls of several plants and collaborating through cloud-hosted applications to serve the demand dynamics. As these cyber physical systems are interfaced with the Internet, they are prone to cyber threats. The necessary criterion for risk assessment in cloud manufacturing is to identify the Internet-enabling interfaces and maintain a database of risks facing them [15–19]. Some of the known cyber security risks to cloud manufacturing systems are the following [12,15–19]:

4.  Eavesdropping attack: an attack mechanism in which the communications from authorized devices to others like them are captured in between by eavesdropping devices (called listeners);
5.  Masquerading attack by capturing packets of unsecured IIoTs: an attack mechanism in which an unauthorized cyber physical system captures packets from unsecured IIoTs and masquerades as an authorized controller to the cloud hosted manufacturing applications;
6.  Distributed Denial of Service (DDoS): an attack mechanism in which massive scale storms of packets are bombarded to the cyber physical systems through unprotected Internet connections compromised by the attackers, thus overloading the computing systems, networking links, and cyber physical controllers;
7.  Side channel attacks: these are penetration attacks through the side channels into the manufacturing network, which are less monitored or ignored by the monitoring systems;
8.  Cross-side scripting attacks: these are Trojan scripts that can be mixed with the running scripts through SQL injection;
9.  Automated code-based attacks (such as Bots): these are penetration attacks caused by pre-programmed automatic codes;
10.  Exploit-based attacks: these are attacks orchestrated through open-source programs created by hacking experts;

11. Identity thefts (of authorized IIoT devices): these are caused by eavesdropping attacks to capture authentication and authorization details of IIoT devices;
12. Insider trading and proliferation: insiders engaged in malicious activities such as injecting Trojan codes in running programs or opening firewall ports for external attackers to launch their exploits;
13. Fake sensor data feeding and actuation attacks in control systems: these are well planned targeted attacks. For example, the attacker may send false signals to lower valve pressure in a key pipeline, thus causing the control system to gradually increase the pressure in the pipeline and lead to an explosion.

The above list provides the answer to the first research question. However, maintaining a database of risks that exposure to the Internet-enabled cloud manufacturing system allows is insufficient for risk management. This is because the controls for protecting the PLCs that have been turned into IIoT devices need to be different from computing systems, keeping in mind their limited processing and storage capacities [17–21]. The cloud manufacturing partners using blockchain technology for smart contracts to execute logistics and supply chain operations can integrate provenance data of IIoT devices with the blockchain [20–22]. Blockchains are comprised of nodes integrated in the form of a chain such that contracts signed for logistics and supply chain operations can be stored in them in the form of encrypted blocks identified and integrated through hash functions. The provenance of computing devices and software systems is a separate database describing the characteristics, ownership, operational modes, and several other such details about those computing devices [23]. Provenance on cloud computing can help in running forensic analysis of past events if recorded separately in addition to the data generated by the events [24]. This can ensure transparency, data fidelity and protection, privacy issues of the data collected, quality control, and intellectual property protection [25]. In cloud manufacturing, the IIoTs and the cyber physical systems enabled by them can be tracked closely using their provenance data [26–28]. Cyber physical systems can be compromised by hackers in several ways. Some known concerns are the following [15–19]:

14. Validating the identity of cyber physical system enabled with IIoT communications;
15. Tracking rapid deployments and Internet-enabling of millions of cyber physical systems;
16. Traceability of cyber physical systems added, modified, and removed, especially installed on mobile assets;
17. Validating fidelity of sensor data sent for influencing process events interpreted out of the sensory data and the decision-making algorithms running the actuation commands;
18. Establishing accountability and liability of individuals owning the cyber physical systems;
19. Inter-cloud assurances of cyber security;
20. Algorithmic transparency (accountability of performance and behaviors of algorithms deployed for controlling operations of cyber physical systems);
21. Cyber physical systems indulging in erroneous or malicious processing, thus affecting the execution of smart contracts negatively.

These six concerns form the problem for which the prototype solution is designed and tested in this research. When manufacturers integrate their cyber physical systems driving their manufacturing processes for cloud manufacturing, they can also integrate their Enterprise Resource Planning (ERP) systems through blockchains [29]. Modern ERPs have interfacing to popular blockchain frameworks such as Hyperledger and Ethereum. In this manner, the IIoTs and their enabled cyber physical systems are also hooked to the blockchain [30,31]. The data collected from them can be stored on big data systems to monitor their events through continuous auditing. By using artificial intelligence, the predictions can be carried out such that actual events versus predicted events can be compared [32]. To understand how this can work, a design scenario of a blockchain-controlled manufacturing network for creating, executing, and monitoring smart contracts using Hyperledger Fabric (based on explanation in Reference [33]) is presented in Figure 1. This design can help visualize where the provenance data streams and artificial intelligence can be positioned in this blockchain.

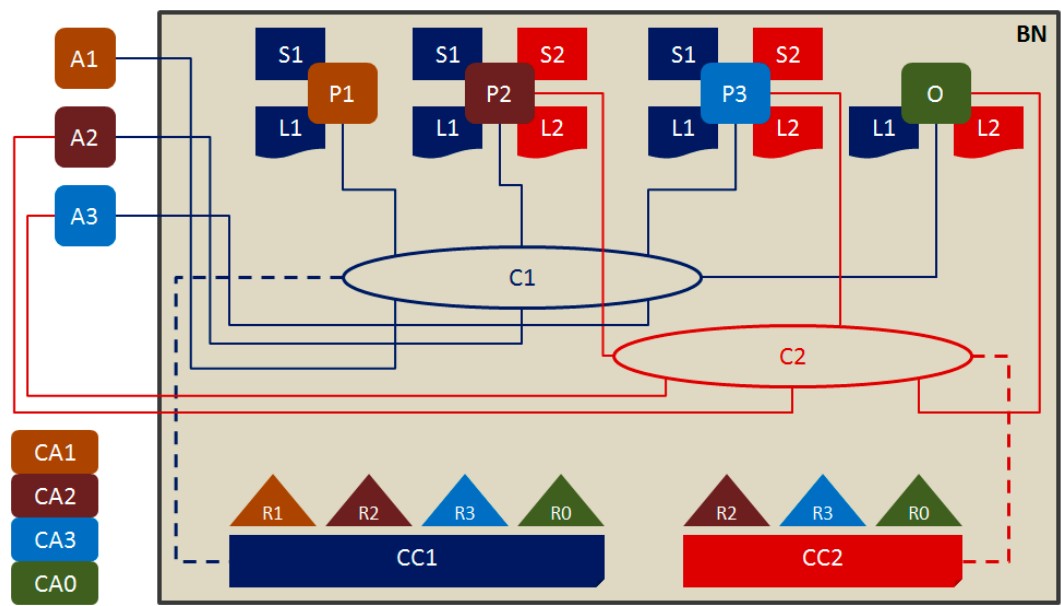

**Figure 1.** A design scenario of a blockchain-controlled manufacturing network for creating, executing, and monitoring smart contracts using Hyperledger Fabric (drawn based on the detailed explanation provided in Reference [33]).

In the scenario shown in Figure 1, four cloud manufacturing organizations (R0, R1, R2, and R3) decide to setup a manufacturing blockchain network (BN) for signing, sharing, and monitoring smart logistics and supply chain contracts. R0 is the contracting authority and others are contracting vendors. They agree to establish two network channels (C1 and C2) governed by policies-based network configurations (CC1 and CC2, respectively). C1 is shared by R0, R1, R2, and R3, and C2 is shared by R0, R2, and R3. Thus, R1 has no business relation with C2 and is therefore denied access to it. A1, A2, and A3 are cloud applications deployed by R1, R2, and R3, respectively, to interact with the network through the authorized channels. To interact with the network, R1, R2, and R3 need to authorize peers P1, P2, and P3 representing them, respectively. R0 authorizes O for managing orders to R1, R2, and R3 through the peers P1, P2, and P3. The peers P1, P2, and P3 are authorized to access the block chain network BN using cryptographic keys issued by certification authorities CA1, CA2, and CA3, respectively. The ordering authority manages C1 and C2 network channels to interact with P1, P2, and P3. P1 has access to C1 only whereas P2 and P3 have access to both C1 and C2. When orders are placed, the smart contracts are signed digitally using signatures issued by CA1, CA2, and CA3 to P1, P2, and P3 for the contracting vendors, and the digital signature issued by CA0 to O for the contracting authority. Further to digitally signing the contracts, CA1, CA2, and CA3 issue X.509 certificates to the components identified as belonging to the organizations R1, R2, and R3, respectively. The certification authorities are also used to sign transactions to affirm their approvals. On signing, the smart contracts are stored in the smart ledgers L1 and L2 belonging to the network segments C1 and C2, respectively. P1 stores a copy of L1 only (as it and its company, R1, do not have any business connection with C2), whereas P2 and P3 store a copy each of L1 and L2. The events related to the smart contracts L1 and L2 (such as approved logs of works completed as per the contractual terms) are stored in state databases S1 and S2, respectively. P1 maintains a copy of S1, and P2 and P3 maintain a copy of S2 and S3 (as per their access rights). All copies of state databases are synchronized. The ordering authority O need not maintain a copy of the state databases because organization R0 is not contributing to state changes. However, O can inspect S1 and S2 at will.

The above design scenario represents a vanilla blockchain network for creating, executing, and monitoring smart contracts and the events linked with their closure. This research added provenance capturing in real time and predictive analytics using artificial

intelligence, for which the random forest algorithm was used. The blockchain framework selected for this research was CORDA [34,35], which is lightweight and can be installed, programmed, and executed in a personal laptop containing an i5 or i7 processor and 16 GB of RAM. The blockchain contract rules were configured in the form of predictive auditing such that the updates are accepted only when the risks predicted by AI are within the prescribed limit.

With massive expansion of cloud manufacturing domains, IIoT devices have proliferated in huge numbers. The security and trustworthiness of IIoT devices are currently not built into the manufactured IIoT devices given their limited computing and storage capacities [17–21]. Hence, the security concerns raised by References [15–19] cannot be addressed by implementing the traditional controls of firewalls, intrusion detection, and web services security as majority of the IIoT devices may not have capacities to implement them. Provenance is an approach to identify, authorize, and authenticate the IIoT devices using secured socket layer and cryptographic keys exchange, which are light weight instruments possible to be implemented in the limited computing capacities of the IIoT devices.

Provenance blockchain on cloud computing and the concept of predictive auditing using artificial intelligence has been studied significantly in the academic world. However, they have been studied as separate themes, providing insights into their applicability for addressing cloud manufacturing cybersecurity risks. The goals of the studies on the two themes are different. Provenance blockchain helps in authenticating and authorizing continuous trustworthiness of IIoT devices, and predictive auditing helps in keeping track of the processes in operation in cloud manufacturing. This research integrates the themes of predictive auditing using artificial intelligence and provenance blockchain. The proposed solution can expand the scope of provenance from authenticating and authorizing the continuous trustworthiness of IIoT devices to complete operational compliance based on rules set by smart contractors in a blockchain dedicated to provenance. Artificial intelligence can help in achieving risk management in real time such that non-complying or even compromised IIoT devices can be detected in advance and long before they could cause actual damage. Thus, the eyes of risks monitoring and control shall not merely be looking at the times when the devices are being inducted, reallocated, and retired, but will be focused on IIoT operations continuously in real time with the power of predictive auditing. Every variable related to IIoT devices allocated to the smart contracts can be monitored and controlled through the proposed solution. The methodology for conducting the primary research is presented in the next section.

## 3. Methodology

This research was conducted with an understanding that cloud manufacturing, which leverages the IIoT devices (cyber-physical systems) for networking over the Internet, might be vulnerable to cyber security risks [9–15]. The cybersecurity systems for IIoT devices need to account for complexities in design and the known limitations of the devices (power, computing, and local storage). The studies [27,28,36,37] clearly highlighted the limitations of capacity and computing power of the IIoT devices, making them inadequate to run sophisticated security programs independently.

The methodology process that follows is comprised of seven steps: study of literature, designing the prototype, selection of software tools, encoding, integration, testing, and evaluation of results.

This research is an original conceptualization of solutions to the concerns related to the employment of cyber physical systems in cloud manufacturing networks, identified by references [15–19]. This research was designed to learn by experiencing a conceptualized design of provenance in the CORDA blockchain framework available with open-source codes. The knowledge was developed through experiencing the coding process and running tests by simulating scenarios of provenance data anomalies using simulated production data collection in logistics processes. The random forest machine learning algorithm was coded in such a way that it could predict numerical values of operating parameters and detect risk levels based on boundary parameters. The risk levels were

appended to the event records confirming work-in-progress and completion as per the terms of the smart contracts. Thus, the operations manager monitoring the event records can view the risk levels and investigate the specific IIoT devices in question. As the events data may be collected from a group of IIoT devices, the entire group may require investigation until a rogue IIoT device is located by the investigators. It requires learning through several rounds of trials and experiences until the results obtained are satisfactory. Keeping this approach in mind, the philosophy selected is pragmatic and the data collection and analysis shall be both qualitative and quantitative [37–39]. The learning was both inductive and deductive. The knowledge gained from literature sources and technical documentation of Hyperledger and CORDA frameworks were qualitative. The knowledge gained from machine learning actions when inbound data are manipulated was quantitative. The accuracy analysis of machine learning was quantitative, as well. Inductive research was required to learn the design and operations of blockchains from the technical documentation of Hyperledger and CORDA frameworks, and programming techniques from the CORDA manuals. Deductive research was required to test the design idea of the researcher by appending the provenance control based on anomaly detection by machine learning. This system can be evaluated by building a prototype in a Linux or Windows environment in a powerful computer and generating data transmission through terminals in the same or other laptops simulating MQTT clients as the IIoT devices. First, a sizeable database needs to be created, comprising 500 to 1000 records, and the artificial intelligence needs to be tested by simulating test cases of IIoT breaches occurring on the manufacturing network. More details on this testing process are provided later in this section.

The state databases S1 and S2 are the main ERP-linked systems receiving regular updates on events completed as per the smart contracts stored in smart ledgers L1 and L2. State changes in the S1 and S2 are facilitated by the ERP applications A1, A2, and A3 on behalf of organizations R1, R2, and R3, respectively, serving R0 through their respective smart contracts. Hence, the responsibility and accountability for the accuracy and integrity of events data being fed to S1 and S2 state databases inside the manufacturing blockchains lies with the administrators of A1, A2, and A3, which are positioned as ERP data systems outside the blockchain. In this research, A1, A2, and A3 are the focal points for building strong provenance security controls. The provenance in this research is not merely static metadata or occasional changes to it; instead it comprises operational and allocation data and rules. In this research, three forklifts are assigned to different physical zones and are assigned to carry different ranges of weights. The location and weight data streams are considered as dynamic provenance feeds.

The proposed modification makes the network a blockchain network with provenance (BNP). The applications A1, A2, and A3 have Message Queuing Telemetry Transport (MQTT) interfaces on which they receive provenance data from IIoT devices attached to the three forklifts. The data from the IIoT devices were used to change the states of state databases inside the blockchain by the blockchain peers P1, P2, and P3. Machine learning (ML) was implemented to make predictions of risk levels by comparing the latest data arriving with their predicted values. The provenance data were stored by ML in a database called ProvDB. The algorithm planned for machine learning was random forest. The reason for using this algorithm is that it is based on supervisory training. As cloud manufacturing provenance data collection is expected to be structured, using supervisory algorithms should be preferred. Other algorithms fitting this space are recurring neural networks, long short-term memory, deep learning, and reinforcement learning [40].

The machine learning shall be trained using historical data in the ProvDB database. To begin a credible learning cycle, around 500 records are planned to begin with in the ProvDB database. The initial records were considered as IIoT inputs from the three forklifts identified as Asset01, Asset02, and Asset03. The random forest was tasked to make asset-wise independent predictions. Hence, it was coded to first export the asset-wise data in separate files and then makes separate and independent predictions about their next state values. The risk levels were calculated by comparing the next state predicted values with

the current state values received. The risks were defined as per the physical boundaries within which the assets were allowed to operate and the loading limit on each asset. The physical boundaries of movements of Asset01 (A01) were: X = 1 to 200, Y = 1 to 200, and Z = 1 to 200 feet, and weight = 100 KG. The physical boundaries of movements of Asset02 (A02) were: X = 201 to 400 feet, Y = 201 to 400 feet, and Z = 201 to 400 feet, and weight = 125 KG. The physical boundaries of movements of Asset03 (A03) were: X = 401 to 600 feet, Y = 401 to 600 feet, and Z = 401 to 600 feet, and weight = 150 KG. In the real world, the location could be a multi-story warehouse in which reach truck forklift machines have been assigned to fixed boundaries. If they breach these boundaries, they will enter the zones of other forklifts and cause accidents. There can also be issues of the wrong forklifts being assigned for jobs not suitable for them (such as capacities of weightlifting). The extent of breach defines the level of risk. For example, if a forklift has breached only the boundary of another forklift, the risk level will be logged as LOW, but if a forklift breaches all the way to the center of the zone of another forklift, the risk level will be logged as HIGH. Four risk levels were assigned: NONE, LOW, MEDIUM, and HIGH.

The state data were entered along with the predictive risks estimation made by the machine learning (ML) by the blockchain peers. Once authorized based on risk levels within acceptable limit, the IIoT devices were trusted to provide genuine events updates from the running processes, which can be used for changing the states in the state databases of the blockchain. However, if the risks are not in the acceptable range, the blockchain state changes would not occur and the peers will be suggested to conduct investigations. The CORDA rules in the blockchain were programmed as follows:

For all assets:

- X should be less than or equal to 600;
- Y should be less than or equal to 600;
- Z should be less than or equal to 600;
- Weight should be less than or equal to 150;
- Risk level should be either NONE or LOW;

To simulate IIoT data transfers, a Message Queuing Telemetry Transport (MQTT) server called Apache ActiveMQ was used. The data in the ProvDB were sent from the Apache ActiveMQ in the form of topic publisher data sent by a publisher file coded as Publisher.Java, which have records matching the database structure of the ProvDB database. The first column is comprised of device keys used for registration and the remaining columns constituted the numerical data collected from the sensors in the running processes. The topic publisher data sent to a subscribed listener code called Listener.java represented the numerical data collected from the sensors. For the purpose of this research, the topic publisher data were constructed manually and sent through the Apache ActiveMQ MQTT Broker server. In real industrial applications, the Publisher.Java shall be embedded as a firmware deployed in physical IIoT devices such that the topic publisher data will be constructed automatically by the industrial sensors integrated in the IIoT devices. For this research, the values are changed manually in the code itself to test different risk levels. The smart contract monitoring can be performed with two quality objectives: the correct forklifts should be assigned to the correct zones and the correct weight loading capacities, and the forklifts should not breach their operating boundaries and enter the zones of other forklifts (unless reallocated operationally).

In this research, deploying real IIoT devices in a laboratory environment was avoided because the study is about detecting anomalies in the data collected from them and not about the engineering of the IIoT devices. The provenance data need to be streamed to the big data systems through highly secured and encrypted channels with appropriate key exchanges, as stated in the studies by References [39–41]. It should be noted that streaming data from IIoTs may not be possible through encrypted links from the devices. IIoT devices are low-capacity low-end systems. Implementing cryptography at the level of cyber physical systems may not be feasible. Hence, chances of breaches are possible.

Provenance data validation is needed in Industry 4.0. This is the value addition proposed and tested in this research.

The topic publisher data constructed manually is comprised of a set of data values tagged to a process at periodic intervals. At every transmission, the values were varied slightly, as is expected in a stable industrial process (up to 10 percent). Intermittently, larger variations were also injected into the data. Machine learning was programmed to learn from the ongoing data streams and predict the next combination of data. A decision rule using the Random Forest algorithm was programmed to compare the predicted versus actual arrival of the next combination of the data. The risks were logged in the form of alerts about four variables: X-axis movement, Y-axis movement, Z-axis movement, and weight lifted. There were boundaries assigned to the four variables. At no breach, the risks were marked as NONE; at one breach the risks were logged as LOW; at two breaches risks were logged as MEDIUM; and at three and above breaches, risks were logged as HIGH. The risks were logged in a log file but were not passed on to the blockchain immediately. Their information was passed onto the state databases of the blockchain only when ten consecutive risk detection events of at least medium level had been logged by the rules engine. The log in the blockchain was not intended to take any automated action but to inform the peers P1, P2, and P3 to begin investigation about specific devices identified as changing their behaviors. The machine learning predictive algorithm was coded using Weka package of JDK 8 and the rules engine were coded using core Java 8 coding.

The framework can be acceptable only if a dynamic range of scenarios is tested within the numerical constraints programmed in the rules engine of the machine learning code. The constraints of physical boundaries and the constraints of weights should be tested in several combinations to generate compliant as well as non-compliant results to the smart contract rules. The behavior of the system can be understood and accepted only if it works satisfactorily under a wide range of dynamic values commensurate with several operating scenarios. Details on the dynamic ranges used for testing are presented in Section 5.

The primary research environment and the tests conducted are reported in the next section. The primary research followed the design of Figure 2 and the scenario explained in this section.

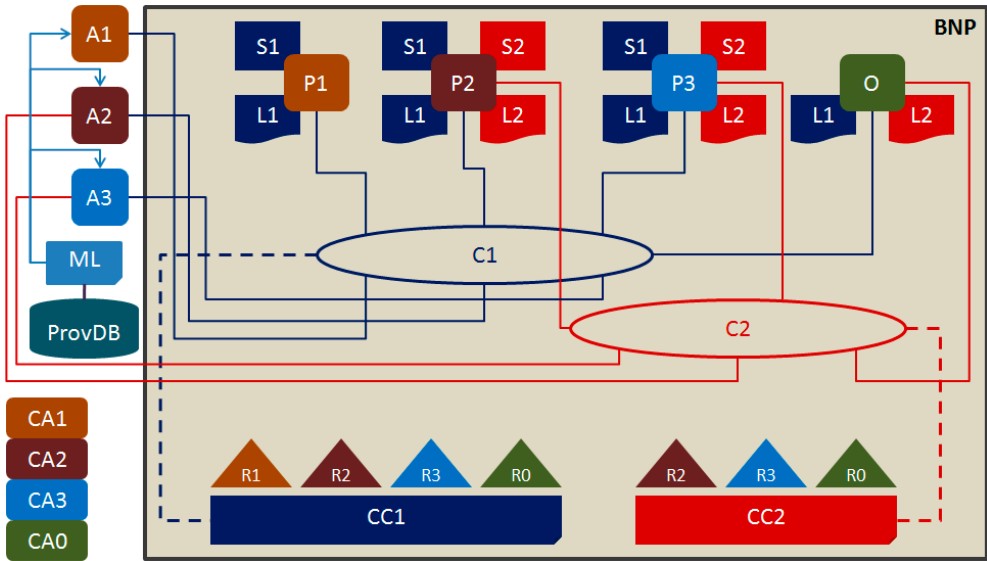

**Figure 2.** Addition of Provenance and Machine Learning to the architecture shown in Figure 1 (Author's own efforts).

## 4. Primary Research

The primary research was conducted by building the software architecture within a laptop environment running Ubuntu 22.04 operating system (a popular distribution of

Linux). The software architecture and runtime flow used for the research project is shown in Figure 3.

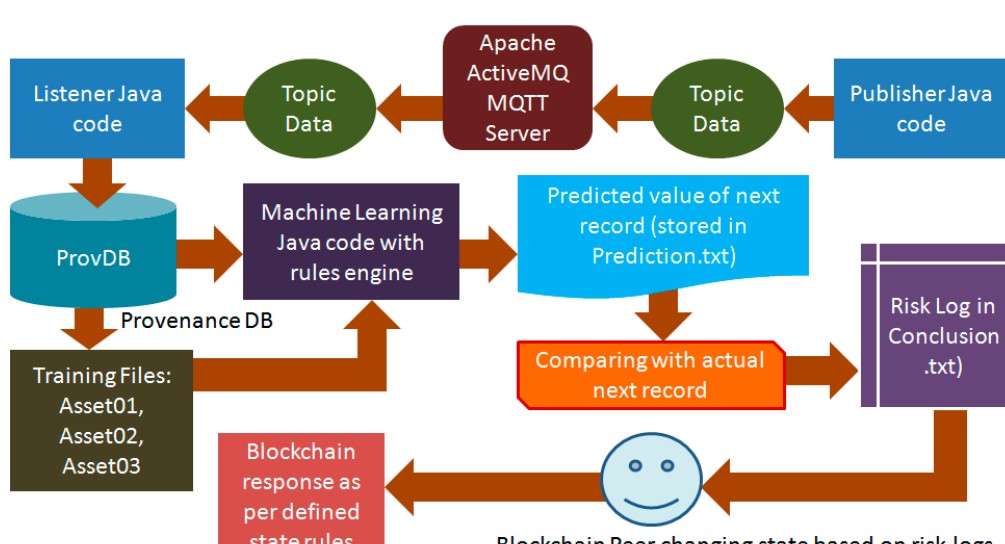

**Figure 3.** Software architecture used for the primary research (Author's own efforts).

Six main components were created and interfaced in the software architecture:

22. MQTT Broker Server using Apache ActiveMQ: The package ActiveMQ-5.15.0 was used to setup a MQTT broker server in Ubuntu 22.04. This package was selected because of its compatibility with Java 8 version.

23. Publisher Java code: Java 8 was selected because CORDA for open-source development is available with packages compatible with up to Java 8 version only. Professional versions of CORDA are available on higher versions of Java. As stated in the previous section, this code should be embedded in the firmware of the IIoT devices. However, for the purposes of testing, this research executed the code several times through Ubuntu 22.04 terminal by manipulating the input data at every event.

24. Listener Java code: programmed using Java 8: this file is responsible for receiving the data transmitted through the MQTT connections and saving them to a ProvDB file to be read by the machine learning algorithm.

25. Provenance database in the ARFF format: The file was created by the Listener java code in ProvDB.csv;

26. Machine Learning code in Java with and in-built rules engine (WEKA): The package selected was weka-3.7.0 because it is compatible with Java 8; the rules or risk assessment and logging were defined as detailed in the previous section. The machine learning code picks up the latest data from the ProvDB and then compares them with the latest next state predictions, using 80% records for training and 20% for testing. Based on the predictions, the machine learning code populates the respective state files of Assets A01, A02, and A03 along with the latest risk values. The verbose report to be analyzed by the blockchain peers is entered in conclusion.txt file in append mode (new records added to the older records).

27. CORDA blockchain framework with state rules defined in Java 8: To create smart contract state rules in CORDA, six Java files were configured: IOUState.java, IOUContract.java, IOUSchemaV1.java, ContractTests.java, ExampleFlow.java, and FlowTests.java. IOU is the name of smart contract tested in this experiment. While configuring these files, a database file named "iou.changelog-v1.xml" is configured automatically. This is the change log database comprising state changes of the smart contract named IOU. All the variables created in Schema and other files such as States and Flow should have an existing record in the change log database.

The tests were run several times following the steps stated below. These steps are the main steps comprised of several technical sub-steps for each.

Step 1: Run the ActiveMQ console;

Step 2: Run the Listener.java file; this opens the MQTT connections in the ActiveMQ console;

Step 3: Run the Publisher.java and transmit data for location coordinates and weight carried out by a Forklift;

Step 4: Run the machinelearning.java file;

The outputs will be generated in ProvDB (Provenance database with the latest record of assetplain.txt appended, asset01.arff, asset02.arff, asset03.arff, output.txt, prediction.txt, and finally, Conclusion.txt;

Step 5: Open the CORDA console through IntelliJ Idea;

Step 6: Try entering the latest values received in assetplain.txt, and analysing the risk log in conclusion.txt and entering the appropriate risk value (updates sent to the smart ledger of the smart contract);

Step 7: Observe the responses from the CORDA smart ledger and write the full report by analyzing it and all the files generated by the machine learning code.

There were some challenges during implementation of the system, though these were solved before running the tests. The Java framework implemented in the laptop was Java 8 because CORDA community edition is currently fully functional and reliable up to this version. Hence, all software frameworks were configured to run on Java 8. The ActiveMQ-5.15.0 package was selected for establishing the MQTT server because it is compatible with Java 8. The other more popular framework for MQTT is RabbitMQ, which was rejected because of incompatibility with Java 8. Further, the weka-3.7.0 package for machine learning was selected for the same reason, although higher versions were available. Implementing them on Ubuntu 22 was not a challenge. These challenges will not appear in the production rollout because the professional edition of CORDA supports higher versions of Java. Further, to keep the sessions alive for MQTT connections during the machine learning session, separate terminals for Publisher.java and Listener.java were kept active while the machine learning code was executed. During this runtime, the laptop's CPU utilisation (four core Intel i5 8th generation CPUs) reaches 100%, reflecting the significant resources required for implementing this solution in practice. Significant hardware resources will be required on cloud computing to run the system stably during such spikes, which is expected to become an almost continuous phenomenon.

Steps 1 to 7 were run for several combinations of input data values about the position and weight carried by three forklifts, identified as Asset01, Asset02, and Asset03 in the blockchain database. The results are discussed in the next section.

## 5. Discussion

The results of all the tests conducted revealed two main states:

28. the machine learning algorithm decides that risk is either at NONE or at LOW level, such that the state change in the CORDA smart contract is allowed;

29. the machine learning algorithm decides that risk is either at MEDIUM or at HIGH level, such that the state change in the CORDA smart contract is prohibited, instructing the blockchain peers to conduct investigations.

One hundred tests were conducted by varying the values following a structured approach. The programming of data concerned a scenario in which three reach truck forklifts (a type of forklifts suitable for high rise warehousing for vertical storage) are allocated to three different operating zones in a warehouse. All the three zones have dimensions of $200 \times 200 \times 200$ feet. They are touching each other but are not interconnected. They were called Zone1, Zone2, and Zone3 in the testing. Zone1 is on the ground, Zone2 is located on a landfill about 200 feet high, and Zone3 is located on an adjacent landfill about 400 feet high. A simple schematic created in Blender 3D software is presented in Figure 4:

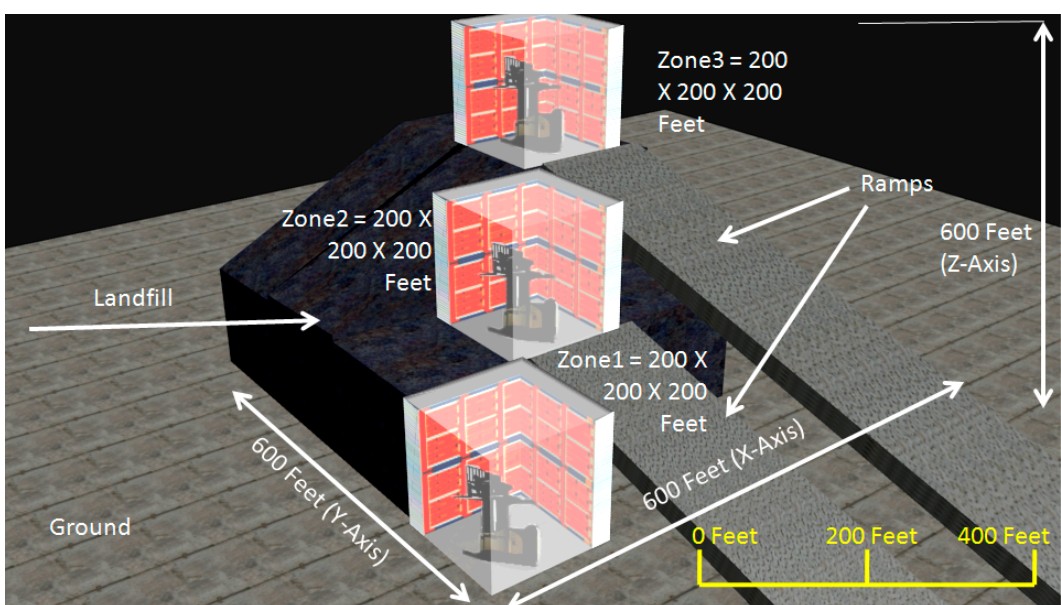

**Figure 4.** Physical layout of the warehouse and the three zones programmed for running the testing steps (author's own design).

As discussed in Section 3, the three reach truck forklifts, named as Asset01, Asset02, and Asset03, were allocated to Zone1, Zone2, and Zone3, respectively. The physical boundaries of movements of the three assets were:

- Asset01 (A01): X = 1 to 200, Y = 1 to 200, and Z = 1 to 200 feet, and weight = 100 kg;
- Asset02 (A02): X = 201 to 400 feet, Y = 201 to 400 feet, and Z = 201 to 400 feet, and weight = 125 kg;
- Asset03 (A02): X = 401 to 600 feet, Y = 401 to 600 feet, and Z = 401 to 600 feet, and weight = 150 kg;

It may be observed that the forklifts can be shifted between zones only when they are removed because Zone2 and Zone3 are accessible through ramps. Hence, breach of boundaries is possible only through planned allocations. The forklifts cannot breach their boundaries on their own. This is the reason this scenario was designed (that is, having no inadvertent breaches unless human actors are involved). The risk levels were defined based on how serious the breach of these constraints was. If a forklift is found to be breaching X or Y, the risk may be LOW because it might have been taken out from the warehouse in a parking place just to give it some rest to cool it down or for some maintenance and repairs. However, if there is a breach in Z-axis constraints (which will occur alongside breaches in the X and Y axes because the forklifts have to be taken out and shifted through ramps), the risk level logged will be MEDIUM to HIGH depending upon how far they have been taken away. The breach in weight levels along with location breach shall generate HIGH risk logging only.

The 100 tests conducted were conducted by entering location data with incremental changes; later even the permissible weights were also breached. Initially, the forklifts were kept within their zones with no breach in weight as well. The risk logs were found as NONE. Thereafter, the forklifts were breached only at X and Y axes by imagining their locations outside their zones but not entering other zones. For example, the forklifts were positioned at various locations on the ground outside Zone1 and on the landfill or the ramps outside Zone2 and Zone3. The risks were found to be logged as LOW. Thereafter, forklifts were entered into other zones and the risk levels were logged as MEDIUM to HIGH depending upon how far they were taken. Finally, when the weights were also breached, the risk levels of HIGH were logged.

In this research, the machine learning code was written in such a way that at every data set received, it conducts prediction of next state values and then compares them with

the data received to log the risks. However, in reality there would never be an abrupt jump to risk levels MEDIUM or HIGH. The transition will be gradual as the assets are moving along their respective paths. This is the reason the tests were conducted by following paths defined in a test scenario. When entering data into blockchain, the blockchain peers should not jump to conclusions. They should follow the trends carefully to see if there are real risks. The use of predictive analytics by AI helps in reducing unnecessary false positives. For example, if the reallocations have happened quite a few times in the past, they will reflect in the predicted values. Thus, the differences between the predicted and actual values will vary significantly only when sudden outliers are caused in the data streams. If a forklift has never been taken to another zone but has been taken out for cooling down or repairs several times in the past, the predictive values will detect a breach only when the Z-value crosses its normal operating range.

The system was found to abide by the rules and produce either of the above two states (a) and (b) without failing on even one of the tests. There were no false positive and false negative risks identified during the tests. However, the actual decision on true positives should be with the blockchain peers as they will compare the risk values with other data available, such as reallocations recorded in the ERP. Normally, the blockchain peer may come across occasional false positives and hence enter NONE and LOW risks in the smart contract state updating. The blockchain peers may agree and establish criteria before they come across real MEDIUM and HIGH risks. As the data flows and risk logs are recorded every second, occasional MEDIUM and HIGH risks can be ignored. For example, they may consider a real MEDIUM risk only after ten or more continuous occurrences, indicating something genuinely wrong with the operations of the specific IIoT asset. Unless a trend is formed and no self-correction is evident, there may be false positives for some reasons (such as a communications failure). If prolonged trends of MEDIUM and HIGH risks are detected, the blockchain peers may correlate them with risk logs of other IIoT assets allocated to the same smart contract and look for causes in the ERP. A quick video conference with the operations team may solve the ambiguity. When they come across real MEDIUM and HIGH risks, they cannot enter them in the smart contract as it will refuse to change the states. In such instances, the blockchain peers will be left with no option but to investigate the risks.

For such responses, documented response and mitigation processes should be implemented showing clear allocation of roles and responsibilities. Perhaps, such processes may be included in the smart contracts.

At this stage, the third research question is answered by explaining how this system can solve the concerns raised by References [15–19] stated in the literature review. This research addresses the concerns, to some extent, as follows:

30. Validating the identity of cyber physical systems enabled with IIoT communications: This concern is clearly addressed in this research as the identity of the cyber physical system is registered in the MQTT broker server, in the ProvDB of the machine learning, in the training and testing database of machine learning, and in the smart contract of the blockchain.

31. Tracking rapid deployments and Internet-enabling of millions of cyber physical systems: With the system of smart contracts and smart ledgers in place, millions of cyber physical systems can be registered and prepared for tracking and tracing before assignment to smart contracts. However, the IT capacities of cloud manufacturing and network bandwidth should be sufficient to handle volumes of data generated by millions of cyber physical systems in real time.

32. Traceability of cyber physical systems added, modified, and removed; especially installed on mobile assets: The machine learning code shall continuously track the operational state changes of the cyber physical systems and log risks accordingly for the blockchain peers monitoring the system. Major changes like addition, modification, and removal cannot go unnoticed by the machine learning code. There may be false

alarms because of communication interruptions (like data received in a topic stops temporarily) but the traceability and tracking will always be ON.

33. Validating fidelity of sensor data sent for influencing process events interpreted out of the sensory data and the decision-making algorithms running the actuation commands: This is a tough challenge. The validity of fidelity of sensor data can only be made by decision-making algorithms comprising of integrated engineering knowledge. This system can, however, report inconsistencies in the state changes through its predictive capability and log risks. This may be of some help to the engineers operating the cyber physical systems.

34. Establishing accountability and liability of individuals owning the cyber physical systems: This concern will be addressed by the system designed. At the time of registration of the assets, the ownership and accountability details will be recorded in the blockchain. The smart contracting parties will be fully liable for the assets registered and allocated to the contract. Further, blockchain peers from all contracting parties will monitor the operations, thus ensuring timely detection of risks logged by the machine learning code.

35. Inter-cloud assurances of cyber security: Inter-cloud assurance is possible in this solution if the same MQTT broker and machine learning systems are implemented for all the contracting parties interfacing through multi-cloud blockchain. If multiple brokers and machine learning systems need to be implemented in a multi-site environment then replication of data between the Conclusion.txt files implemented in multiple clouds should be implemented.

36. Algorithmic transparency (accountability of performance and behaviors of algorithms deployed for controlling operations of cyber physical systems): This is another tough challenge to be addressed. Performance and behaviors of algorithms require much deeper monitoring and control by sophisticated systems with full knowledge about the operating behaviors and performance metrics of the algorithms. This system can help by detecting changes in the already progressing patterns and reporting them as risks at different levels depending upon the rules defined in the machine learning code and the blockchain state rules.

37. Cyber physical systems indulging into erroneous or malicious processing using exploits, scripting attacks, bots, device identity theft, and other means thus affecting the execution of smart contracts negatively: Detection of exploits, scripting attacks, bots, etc., need to be enabled through intrusion detection and prevention systems. This solution can detect operational anomalies caused by such malicious software attacks through machine learning but cannot detect presence of the software. Any anomaly causing negative execution of smart contract will be detected through machine learning and traced to the device using provenance data. Negative execution will breach the blockchain state transition rules and hence transactions will be rejected, thereby promoting investigations. Attempts of device identity theft will also be difficult for the attackers because three levels of registration in MQTT broker, machine learning ProvDB database, and the blockchain smart contract will cause deterrence for the attack planners. There is a high chance that the attackers will not be able to plan a perfect breach of this entire system, although they should never be underestimated.

The challenges for provenance verification system to identify the IIoT devices accurately and build traceability of doubtful devices in the network can be addressed. Further, the challenges of provenance detection of bindings, fault tolerance, integrity and confidentiality verifications through data, chain, and origin integrity verifications, access controls, and protection of keys during sharing may be solved to some extent following the solution of continuous operational risks monitoring in this research. Once devices are registered in the blockchain, they will be treated as trustworthy in the system tested in this research. However, this will not be a permanent perception built about the devices even if they follow all the routines and key exchanges for valid registration. Devices may be subject to investigation if their operational boundaries are breached and risks from medium on-

wards are logged because the blockchain state change will not be allowed. The next state prediction will always be based on the historical gradual state changes and hence any drastic variations will be detected promptly. Further, the predictability can also cover chances of devices breaching their operational boundaries in the normal course of their operations. As the blockchain peers are monitoring the records periodically and updating state changes in the blockchain, they will have opportunities to detect such probabilities well in advance and correct the course of operations of the devices to mitigate such risks proactively. There may be chances of some false positives as the devices may have been reallocated deliberately through mutual agreements among the blockchain contracting parties. However, reallocations should always be conducted through new smart contracts such that the MQTT broker server and the machine learning rules can be updated.

In real world implementation, there may be thousands of IIoT devices allocated to smart contracts running their work-in-progress operations. The data streams from these IIoT devices to the MQTT server may establish and disestablish thousands of connections every second. The machine learning code will be required to launch training and test files from ProvDB corresponding to every IIoT device separately for making individual predictions and comparing with their respective smart contract rules engines. There may be thousands of such files launched in real time, operating almost synchronously with the MQTT connections as the latest next state data of every IIoT asset will be compared with the latest predicted data for detecting compliance/breaches of smart contract rules. Keeping these predictions in mind, one may imagine a massive scale cloud computing system in action, integrated with cloud manufacturing. Perhaps virtual machine clusters comprising hundreds of multi-regional virtual machines will be required. The designers need to consider estimates of overheads and linked costs for adopting this system. The feasibility of investments can be justified when compared with the costs of IIoT breaches and related incidents suffered by cloud manufacturers. Another factor related to adopting the system in practice to be considered is the latency. The machine learning predictions need to be synchronised with the data streams from thousands of IIoT devices. The risk logs need to be created in real time on arrival of the present state data. The network connections need to be configured with sufficient bandwidth and the distances covered by the IIoT traffic to the provenance databases on cloud manufacturing should be the shortest open paths. Finally, it is suggested that the proposed system can be used for multiple contexts of cloud manufacturing in the Industry 4.0 era for future research or real world adoption. This research presented coded files for managing provenance data streams and related machine learning for forklifts programmed to work under physical boundaries and weightlifting constraints as per the smart contracts. The application can be modified by adding a frontend providing control on various contexts related to the IIoT devices connected with cloud manufacturing and ready to be allocated to smart contracts. The application shall comprise rules generation through a frontend such that the contract generators can define rules related to the variables associated with IIoT devices selected for the smart contracts. The frontend should be able to generate the necessary entries in the machine learning code at the backend. With such a generic system in place, the machine learning code can be used for monitoring the provenance streams and identifying risks related to different types of IIoT devices connected to a cloud manufacturing network. The addition of new IIoT devices shall be conducted by defining their variables pursuant to their sensing capabilities such that they can be governed by smart contract rules generated through the generic frontend used by the contract generators.

Several research studies [41–44] established designs requiring key exchanges with client systems and their validation by their authorization records stored in the provenance blockchain. The threats to be addressed were covered by references [15–19]. This research was conducted with a basic understanding that the key exchange and validation mechanisms suggested by References [41–44] can address identification, authentication, and authorization of the IIoT devices. However, several IIoT issues highlighted by References [15–19] shall remain unaddressed. The proposed system was designed and tested for

continuous validation using artificial intelligence. In this context, the justifications of the proposed system for each of the threats identified by the References [15–19] were included earlier in this section. This research also addresses the concern of algorithmic transparency raised by References [13,14]. As the artificial intelligence operates by monitoring the variables associated with the IIoT devices directly, any differences between the algorithmic operations and the smart contract terms for provenance of the devices shall be detected automatically through the risk logs. The issue of controlling uncertainties in modern cloud manufacturing raised by References [4,5] may be addressed through the proposed system by monitoring the IIoT variables closely in real time. There will be implications of costs and overheads, though, that need to be addressed by the planners and designers.

In the larger framework of cybersecurity, the proposed solution of provenance blockchain can gain a prominent role of activity logging, monitoring, and control in the SIEM (Security Informatics and Events Management) framework [45]. The SIEM framework is relevant for real time visualization of events occurring in critical manufacturing systems [46]. The provenance blockchain is proposed to be useful for the SIEM framework as the activities of IIoT devices can be monitored and compared with their pre-established operating boundaries in real time. The blockchain peers can be empowered by the continuous flow of monitoring of states and knowledge of breaches to ensure that exploited vulnerable IIoT devices can be identified and administered quickly before major damages occur.

Cloud based manufacturing systems may be able to reduce or eliminate the risks arising from cyber physical system security challenges if they are monitored and controlled continuously in real time. Disasters in industrial systems may require a build-up period before they actually occur. For example, an explosion in a boiler will occur after prolonged building of pressures beyond their operating boundaries. The proposed provenance blockchain solution may be able to detect such build-up sequences during their build-up periods and the blockchain peers may be able to take timely protective and preventive actions. However, it will not happen automatically, as intelligent interpretations and prompt actions taken by the blockchain peers is required for the solution to work. Further, the issues of performance overheads caused by the provenance blockchain, scalability to hundreds of thousands of IIoT devices, and latency issues causing delays in flow of state data from the IIoT devices need to be addressed. Future researchers may like to conduct quantitative evaluation of these issues.

IIoT sensing streams used as provenance data validated by artificial intelligence for making state changes in smart contracts stored in blockchains can have several business benefits. The smart contract state rules can be defined to enforce any policies on the whole cloud manufacturing network. In future, this model and its design and coding may be useful not only for cyber security risk mitigation but also for increasing efficiency and productivity, an increase in trust and transparency in the manufacturing process, and promoting sustainability. For example, if the contract demands emission levels from logistics and transportation equipment to be below defined thresholds, the IIoT data sensed and the machine learning driven risk levels thus logged can be useful in accepting or rejecting state changes in the smart contracts putting compliance pressure.

## 6. Conclusions

This research was conceptualized with three research questions replicated as the following:

38. What are the risks associated with cloud manufacturing in Industry 4.0?
39. How can provenance blockchain be used to provide greater transparency and traceability in the cloud manufacturing process using AI-enabled predictive auditing?
40. How can this system help in mitigating cloud manufacturing risks in Industry 4.0?

In response to the first research question, the cyber security threats to cloud manufacturing were listed based on review of literature. Sophisticated threats like code injections, side channel attacks, covert channels, exploits, malware, and DDoS may be mitigated on cloud computing because of high end security controls. However, cloud manufacturing

is dependent upon the data streams from IIoT devices attached with the process event sensors in the plant machinery, robots, and logistics equipment. They may not be protected from the sophisticated threats because of low computing and storage power requiring use of low end thin operating systems (such as Lubuntu, which is a lightweight version of Ubuntu). If IIoT devices are compromised especially by insiders, they can be used as launch pads for attacking the manufacturing infrastructure. The concerns related to IIoT devices were identified separately, which were related to cyber security and beyond related to operational reliability, trust, and quality assurance. Their behavioral trends need to be monitored to find out if they are compromised making them rogue devices. A promising solution evolving in scientific research is using provenance blockchain employing predictive capabilities of AI. This concept was adopted in this research as the second research question. The second research question was answered by studying the provenance, blockchain, and AI solutions for cloud manufacturing as separate themes, which were combined in a design realized within an Ubuntu laptop environment. A scenario was imagined in which, three reach truck forklifts allocated to three separate zones in a warehouse having constraints in the form of physical operating boundaries and weights.

AI was programmed to detect breaches to the constraints and logging the risks using random forest predictive analysis and an in-built rules engine within the coding. Thereafter, a smart contract system was programmed in CORDA framework including the risks in the state change rules for tracking the events conducted to fulfill the smart contract's requirements. Provenance blockchain can be deployed for monitoring provenance data flows such that smart contracts can be defined with operating rules for the IIoT enabled devices allocated to processes to be executed under smart contracts. Predictive auditing can be integrated with cloud manufacturing process using AI algorithms to predict operational behaviors of the allocated devices and log risks visible to all blockchain peers and the contracting parties. This system is expected to increase transparency and traceability in the manufacturing process. The provenance logs can highlight the devices with the best compliance history and in this process highlight the contractors able to ensure best trustworthiness in device allocation and management. Long term records can be used to measure the performances of all the devices registered with the blockchain. Transparency and traceability were ensured by making the event logs, predictive AI results, and the risk logs available to all blockchain peers and the customer.

With this system in place, the third research question was framed to explore how it can help in mitigating the cloud manufacturing risks. To answer this question, the AI power was added to provenance blockchain to improve risk cloud manufacturing process as it will be accurate, timely, and automated. The AI was designed to append the Provenance database and segregate it to generate device-wise training files such that device-wise predictions of their next operational attributes could be generated and compared with the latest state to detect operational breaches and log risks in the risk database. This risk database is visible to all blockchain peers. Hence, they were tasked to conduct periodic risk assessment and enter their reports in the smart ledger. The smart ledger would reject state changes at higher risks, thus putting pressure on the blockchain peers to escalate and investigate the risks proactively.

Several tests were conducted to deeply experience the behavior of this system. The layout of the warehouse imagined for this research was drawn in Blender 3D software and presented for visualizing the risky scenarios. Thereafter, the possible risky scenarios were discussed. As this system operates in real time, risk logs happen at each data transmission event and data comparison between AI predictions and actual. Using AI predictions shall reduce the chances of false positives as the risk levels will increase gradually by following the paths of the vehicles on their way to breaching the boundaries. However, the blockchain peers need to correlate the risk values with all other data available in the ERP systems. If they detect deliberate human actions logged in the system officially, then they can safely assume the risks to be in control and update the events in the blockchain smart contracts. However, if the human actions are found to be not declared in ERP officially, then they can

delay updating the state changes and first investigate the reasons. It was visualized that such real time monitoring of risks can help in risk mitigation to any IIoT related risks, such as quality risks, reliability risks, sustainability risks, efficiency risks, productivity risks, and any other area of concern of the engineers. The right kind of IIoTs and sensors need to be selected, and the Java rules defined for risk assessment need to be customized as per the variables being monitoring. The random forest algorithm will simply predict new numbers based on its training and testing data, and the rules engine shall detect and publish the associated risk levels and their related actions.

**Author Contributions:** M.A.U.: Conceptualization, Methodology, Formal Analysis, Investigation, Validation, Software, Visualization, Writing–original draft. E.G.B.: Supervision, review and editing. L.B.G.: Supervision and reviewing. All authors have read and agreed to the published version of the manuscript.

**Funding:** This research received no external funding.

**Data Availability Statement:** Data are contained within the article.

**Acknowledgments:** This research acknowledges the support of Addis Ababa Institute of Technology (AAiT), for access to library and other resources used in the studies.

**Conflicts of Interest:** The authors declare no conflicts of interest.

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
