# Peer review of "Leveraging Artificial Intelligence and Provenance Blockchain Framework to Mitigate Risks in Cloud Manufacturing in Industry 4.0"

_electronics, doi:10.3390/electronics13030660_

Round 1

Reviewer 1 Report

Comments and Suggestions for Authors

This article explores collaborative cloud manufacturing in Industry 4.0 and its associated risks. The manuscript is well-written, and the findings hold significance for the academic community. Following a comprehensive literature review, the authors delve into the realms of provenance, blockchain, and AI solutions for cloud manufacturing, amalgamating them into a cohesive design executed within an Ubuntu laptop environment. Subsequently, a hypothetical scenario is devised, involving three reach truck forklifts assigned to distinct zones within a warehouse, each governed by physical operating boundaries and weight constraints. The AI system is intricately programmed to identify breaches to these constraints, logging associated risks through random forest predictive analysis and an embedded rules engine in the code. The ensuing step involves the development of a smart contract system within the CORDA framework, encompassing the risks in the state change rules to monitor events fulfilling the smart contract's requisites.

Given the lucid presentation of results and the alignment of the topic with the journal's scope, I recommend publication with minor revisions:

-       Incorporate a dedicated scale for Figure 4.

Reviewer 2 Report

Comments and Suggestions for Authors

1.  The authors need to establish the causal relationship between the assertion that "provenance blockchain framework" can address the risk arised from the cloud based manufacturing environment.  Are those dimensions addressed on page 13 necessary and sufficient to address the risk?

2.  Section 1: The authors should clarified the kind of risks unique to the cloud manufacturing vs. general manufacturing environment.  

3.  Section 2:  One of fundamental challenge in the manufacturing environment is about the tolerable latency and maximum  power consumption (especially for IIoT environment)  This might limit the choice of blockchain approach - especially when zero knowledge proof approaches.  The authors need to discuss the trade off here.

4. Section 2: What are the necessary and sufficient criteria for considering risk mitigation for cloud based manufacturing environment?

5. Section 3: The methodology should clarify what constitute acceptable framework.  Does the methodology needs to be validated through static and dynamic testing in a cyber range?

6.Section 4: This environment (in Fig 3) might be too over-simplified compared to a genuine manufacturing environment.  This section needs to be redone to define a reasonable and representative environment so that the simulation can be convincing.

7. Section 5: There are only qualitative analysis and no baseline.  This should be extended to include a baseline as well as quantative analysis.

Comments on the Quality of English Language

None

Reviewer 3 Report

Comments and Suggestions for Authors

This is an intriguing topic as it combines two of the most recent and impactful topics of academic research: artificial intelligence and blockchain techniques. However, I believe that this qualifies more like a case study and less as empirical research. The paper has a sound structure and touches on a few interesting academic points, such as Industrial Internet of Things (IIoT) devices, cloud computing, 14 Internet communication, and big data analytics. However, its scope is related to physical boundaries and weight lifting limits allocated to forklifts and their respective mechanisms of detecting risks of breaking these constraints with the help of artificial intelligence. Although the paper displays the structure of academic research, including research questions (which largely remain unanswered), literature review, methodology, and discussions, all these basically focus on a case study, with little avenue for a theoretical contribution. However, it captures some useful insights that can be considered useful for the future. My advice for the authors would be to include an empirical study as part of their framework, use some form of scientific methodology to prove their constructs and try to approach the entire process from a theoretical perspective.

Comments on the Quality of English Language

The quality of the English phrasing is fine, although a bit cumbersome and hard to read at some points. However, this does not significantly impact the quality of the paper.

Reviewer 4 Report

Comments and Suggestions for Authors

The paper investigates the application of a customized provenance design within the CORDA blockchain framework, coupled with a random forest machine learning algorithm. The goal is to address risks associated with cyber-physical systems in cloud manufacturing networks. The study employs both qualitative and quantitative methods, leveraging blockchain, machine learning, and provenance to enhance security controls. The proposed Blockchain with Provenance (BNP) modification integrates MQTT interfaces, machine learning, and smart contracts to manage risk levels in a cloud manufacturing environment. However, I suggest enhancing the quality of the paper as follows:

  1. 1- Provide more details on the challenges faced during the implementation of the proposed system, especially concerning the integration of MQTT interfaces, machine learning, and blockchain. Discuss any limitations encountered and potential solutions.

  2.  
  3. 2- Consider expanding on the scalability aspects of the proposed solution. How well does the system handle an increasing number of IIoT devices and events in a real-world scenario?

  4.  
  5. 3- Clarify the decision-making process for blockchain peers when encountering false positives. How are conflicts resolved, and what mechanisms are in place to ensure accurate risk assessments?

  6.  
  7. 4- Discuss the potential overhead or latency introduced by the integration of machine learning algorithms into the cloud manufacturing network. How does the system handle real-time data processing requirements?

  8.  
  9. 5- Explore the applicability of the proposed solution to different Industry 4.0 contexts or specific manufacturing domains. Are there modifications needed for diverse use cases, and if so, what considerations should be taken into account?

  10.  
  11. 6- Consider discussing the economic feasibility and resource requirements for implementing the proposed system in a real-world manufacturing environment. How cost-effective is the solution, and what infrastructure is necessary?

Comments on the Quality of English Language

N/A

Reviewer 5 Report

Comments and Suggestions for Authors

The writing and format has to be improved. I listed some examples here.

a. "Authors should discuss the results and how they can be interpreted from the per- 467

spective of previous studies and of the working hypotheses. The findings and their impli- 468

cations should be discussed in the broadest context possible. Future research directions 469

may also be highlighted." This sentence should be deleted.

b. The numbering of " 1.2.3..." in page 3 and 14 should be like" 1), 2), 3)..."

c. Not quite sure the meaning of "this goal" in the abstract.

d. "A software 22

architecture comprising IIoT communications to machine learning for comparing the latest data 23

with the predictive auditing outcomes and logging appropriate risks..." This sentence is too long. 

e. Space (abstract section, line 30) should be deleted.

f. Using " This research" twice in abstract. It can be like " this work". 

g. "As this is an original concep- 275

tualisation to solve the research problem.." This sentence should be revised. 

Comments on the Quality of English Language

The writing and format has to be improved. I listed some examples here.

a. "Authors should discuss the results and how they can be interpreted from the per- 467

spective of previous studies and of the working hypotheses. The findings and their impli- 468

cations should be discussed in the broadest context possible. Future research directions 469

may also be highlighted." This sentence should be deleted.

b. The numbering of " 1.2.3..." in page 3 and 14 should be like" 1), 2), 3)..."

c. Not quite sure the meaning of "this goal" in the abstract.

d. "A software 22

architecture comprising IIoT communications to machine learning for comparing the latest data 23

with the predictive auditing outcomes and logging appropriate risks..." This sentence is too long. 

e. Space (abstract section, line 30) should be deleted.

f. Using " This research" twice in abstract. It can be like " this work". 

g. "As this is an original concep- 275

tualisation to solve the research problem.." This sentence should be revised. 

Reviewer 6 Report

Comments and Suggestions for Authors

Referee report

This paper explores a practical approach to achieve research goal. This model has the potential to enhance efficiency and productivity, foster greater trust and transparency in the manufacturing process.  Therefore, I think that the paper makes a contribution and has the potential to be published.  However, I summarize in the GENERAL COMMENTS as follows:

GENERAL COMMENTS

1. In Literature Review section, the literature review section should provide an analysis and summary to emphasize the significance of this study.

2. In Methodology section, the section should emphasize the methodology process.

3. On lines 311-312, the machine learning (ML) was implemented to make predictions of risk levels by comparing the latest data arrived with their predicted values. There are many machine learning methods. Here, machine learning was random forest. Please provide the reason for using this method. Are other machine methods also applicable to this system?

4. How to evaluate this system?

5. The results are not significant enough for journal publication yet. More comprehensive evaluations are needed for journal publication.

Round 2

Reviewer 2 Report

Comments and Suggestions for Authors

The presentation quality of the revised paper has significantly improved.  Nevertheless, the authors are recommended to clearly present the reasoning process for the proposed contributions:

1. cloud manufacturing (which might leverage IoT devices) might be vulnerable to new cybersecurity risks.  

2.  cyber physical system security call for solutions to address challenges in power and computing complexity that can be contained inside IoT or IIoT systems.

3.  this area has been intensely surveyed during recent few years, and a number of standards have been proposed through various government agencies (including NIST, DHS in US) and a number of professional socieities (such as SAE G-32).

4. the authors are encouraged to leverage the overall cybers physical ecurity framework needing to be addressed and how the proposed blockchain provenance framework fit within the overall framework.  

5. The authors should also be able to answer the question in the final section: with the help of the proposed blockchain provenance framework, will cloud based manufacturing system be able to reduce or eliminate the risks arising from cyber physical system security challenges. 

Comments on the Quality of English Language

The English of the manuscript seems to be largely ok.

Reviewer 4 Report

Comments and Suggestions for Authors

I'd like to thank the authors for their feedback. However, I believe they need to address the issues raised by performance comparison, such as potential overhead, latency, and scalability in quantitative methods.

Comments on the Quality of English Language

N/A

Reviewer 6 Report

Comments and Suggestions for Authors

Referee report

This paper explores a practical approach to achieve research goal. This model has the potential to enhance efficiency and productivity, foster greater trust and transparency in the manufacturing process.  The authors carefully revised the manuscript and made some changes to the version according to the comments of the reviews. Therefore, I think that the paper makes a contribution and has the potential to be published.

Round 3

Reviewer 2 Report

Comments and Suggestions for Authors

Revision did not seem to fully address all of the concerns raised in the review report.  No references given in the added paragraphs.   

Comments on the Quality of English Language

Quality of English is largely ok.
